# Deep brain activities can be detected with magnetoencephalography

F. Pizzo [1,2], N. Roehri[1], S. Medina Villalon[1,2], A. Trébuchon[1,2], S. Chen[1], S. Lagarde[1,2], R. Carron[1,3], M. Gavaret[4], B. Giusiano[1], A. McGonigal[1,2], F. Bartolomei[1,2], J.M. Badier[1] & C.G. Bénar[1]

The hippocampus and amygdala are key brain structures of the medial temporal lobe, involved in cognitive and emotional processes as well as pathological states such as epilepsy. Despite their importance, it is still unclear whether their neural activity can be recorded non-invasively. Here, using simultaneous intracerebral and magnetoencephalography (MEG) recordings in patients with focal drug-resistant epilepsy, we demonstrate a direct contribution of amygdala and hippocampal activity to surface MEG recordings. In particular, a method of blind source separation, independent component analysis, enabled activity arising from large neocortical networks to be disentangled from that of deeper structures, whose amplitude at the surface was small but significant. This finding is highly relevant for our understanding of hippocampal and amygdala brain activity as it implies that their activity could potentially be measured non-invasively.

[1] Aix Marseille Univ, INSERM, INS, Inst Neurosci Syst, Marseille 13005, France. [2] APHM, Timone Hospital, Epileptology and cerebral rhythmology, Marseille 13005, France. [3] APHM, Timone Hospital, Functional and Stereotactic Neurosurgery, Marseille 13005, France. [4] INSERM UMR894, Paris Descartes university, GHU Paris Psychiatrie Neurosciences, 75013 Paris, France. Correspondence and requests for materials should be addressed to F.P. (email: francesca.pizzo@ap-hm.fr) or to C.G.Bén. (email: christian.benar@univ-amu.fr)

Hippocampus and amygdala are two brain structures situated in the mesial part of the temporal lobe. They play a key role in a plethora of physiological and pathological mechanisms, such as memory organization[1], emotion regulation[2], and epileptogenicity[3]. Because of their deep localization, and their "closed field" configuration[4], it is challenging to record their activity by means of surface measurements[5]. Thus, currently, intracranial electrode implantation, in particular stereoelectroencephalography (SEEG)[6], is the gold standard to access these brain structures[7]. However, SEEG is an invasive surgical procedure, performed under general anesthesia, which requires a complex and multiphase process, a high degree of medical expertize[8] and may rarely be associated with hemorrhagic or infectious complications. These inherent constrains preclude wide application of this method outside of the domain of pre-surgical evaluation of focal drug-resistant epilepsy. The possibility of non-invasively recording amygdalar and hippocampal signals using surface methods such as magnetoencephalography (MEG) and electroencephalography (EEG) could therefore be very attractive. EEG and MEG techniques are readily applicable to both normal subjects and patients and could dramatically improve our knowledge about the functions and dysfunctions of mesial structures.

Computational modeling studies suggested that mesial structures, such as amygdala and hippocampus, produce a signal that could be non-invasively recorded at the surface applying appropriate protocols[9,10] and techniques[11]. However, empirical evidence of such detectability is still debated.

So far, evidences of mesial brain structures surface detectability have been mainly obtained by evoking activity in hippocampus[12–14] or amygdala[15–17] through-specific experimental protocols supposed to activate these structures. To non-invasively record spontaneous activity is more difficult, as it is not time-locked and generally low in amplitude. With the aim of recording spontaneous amygdala and hippocampus activity, temporal lobe epilepsy (TLE) has been largely used as a model. The pathological hypersynchronous epileptiform discharges in the form of "interictal spikes" which are generated in these two structures in patients affected by TLE, represent in fact a potentially good trigger for their non-invasive localization.

An issue with TLE is that amygdala and hippocampus are often activated within a larger network involving other neocortical structures, as evidenced by SEEG[18,19]. It is thus very difficult to ensure that the observed surface signals effectively come from the hippocampus or amygdala and not from the nearby neocortical structures. Data published to date in epilepsy patients have yielded conflicting results, both proving[20–23] or disproving[5,24–26] their recordability from surface measurements. One main constraint is indeed the difficulty in distinguish signals coming from amygdala and hippocampus from those coming from nearby neocortical structures.

Independent component analysis (ICA)[27–29] which allows decomposition of a multivariate signal into independent subcomponents, separating signals arising simultaneously from different sources, could be especially effective in solving such a problem. In previous work from our group[29], ICA on MEG recordings was shown to be useful in differentiating interictal epileptic networks from other surface signals, and the accuracy of such a distinction was confirmed by SEEG.

A method of choice for validating the results of surface ICA is the simultaneous recording of intracranial and surface signals, where SEEG represents a "ground truth" obtained directly within deep brain structures, with anatomical precision of electrode contact localization at the millimeter scale confirmed by magnetic resonance imaging (MRI).

Due to the technical difficulties in the settings of combined recordings, very few studies with simultaneous intracranial and surface recordings[21,30–32] have been conducted so far, often presenting scarce intracerebral sampling[33,34]. Using a method developed in our laboratory[35], we are now able to record simultaneous SEEG-MEG signals with high SEEG sampling and high signal quality[36].

In the present work, we hypothesized that a contribution of hippocampus and amygdala at a MEG sensors level exists but is hidden by neocortical activities. We used ICA as a way to discriminate signals originating concurrently from different parts of the epileptic network, thus providing an automatic separation of concomitant events. To validate our hypothesis, we simultaneously recorded SEEG and MEG in patients with focal drug-resistant epilepsy, applied ICA analysis on MEG recordings and subsequently validated the surface MEG results by calculating the correlation with the simultaneous intracranial signal[35]. Furthermore, we performed source analysis of the ICA topographies in order to test their mesial origin.

## Results

**Mesial structures are visible on MEG using ICA triggered by SEEG.** To evaluate detectability of hippocampus and amygdala on MEG, we performed ICA on MEG signals triggered by SEEG hippocampus or amygdala spikes as previously marked on the simultaneous intracranial recording ("SEEG-triggered analysis"). This analysis was intended to test the "forward problem" of visibility of mesial structures on MEG.

Across patients, between 16 and 185 spikes were marked visually on the SEEG traces, either in the amygdala or hippocampus. ICA on MEG signals resulted in a series of components, each having a spatial part (topography on the sensor) and a corresponding time course. To probe the relationships between ICA components and SEEG signals we used two automatic complementary methods: correlation across time (on concatenated spikes) and inter-trial correlation (at each time point across spikes). Moreover, to ensure that the time courses were effectively correlated at zero lag as evidenced by the two automatic analyses, we further performed visual inspection of the averaged time courses of the ICA components and the SEEG channels they correlated with. We thus verified that the peaks of activity on MEG-ICA and SEEG were temporally aligned. This increased our confidence that we did not observe delayed signals on MEG that could correspond to propagated activity. The components that had significant correlations with the same structures in both tests and that were subsequently confirmed by visual analysis were retained.

Moreover, to evaluate whether MEG mesial ICA components could be localized from surface recordings ("inverse problem"), we applied source localization to the ICA topographies. We then compared the results of source localization with the SEEG contacts presenting significant correlation with MEG.

Using SEEG-triggered analysis, eight patients out of 14 showed at least one significant correlation between one ICA component and a mesio-temporal structure (Table 1, Supplementary Data 1). Among these patients, we found in four patients (P1, P3, P7, P9) significant correlation with hippocampus alone, and in three other patients (P2, P4, P12) significant correlation with hippocampus together with other mesio-temporal structures (collateral sulcus, parahippocampal gyrus, fusiform gyrus, or amygdala). For one patient (P5) we found significant correlation with amygdala alone (Fig. 1).

Source localization analysis, when applicable (9/11 components in seven patients with a high goodness of fit—see Methods section), was able to confirm the mesial origin of the components

**Table 1 Mesial ICA characteristics.**

| | SEEG marker position | Lat | SEEG-triggered analysis-mesiotemporal ICA | SEEG-triggered analysis-thalamic ICA | SEEG-triggered analysis-structure correlated | SEEG-triggered analysis-source localization | Continuous analysis-mesiotemporal ICA | Continuous analysis-thalamic ICA | Continuous analysis-structure correlated | Continuous analysis-source localization |
|---|---|---|---|---|---|---|---|---|---|---|
| P1 | Hippocampus | L | Yes | None | Hippocampus | Hippocampus | Yes | None | Hippocampus | None |
| P2 | Hippocampus | R | Yes | None | Hippocampus and parahippocampal gyrus | None | Yes | None | Hippocampus; Hippocampus and parahippocampal gyrus | Hippocampus |
| P3 | Hippocampus | L | Yes | None | Hippocampus; Perirhinal cortex and collateral sulcus; Hippocampus and parahippocampal gyrus | Perirhinal cortex and collateral sulcus | Yes | None | Hippocampus; Hippocampus and perirhinal cortex or parahippocampal gyrus; collateral sulcus; perirhinal cortex | Hippocampus, Perirhinal cortex |
| P4 | Hippocampus | R | None | None | None | None | Yes | None | Hippocampus; Collateral sulcus | Mesial not concordant (thalamic) |
|  | Hippocampus | L | Yes | None | Hippocampus, parahippocampal gyrus, collateral sulcus and fusiform gyrus | Hippocampus, parahippocampal gyrus, collateral sulcus and fusiform gyrus | Yes | None | Amygdala; Perirhinal cortex | None |
| P5 | Hippocampus | R | Yes | None | Amygdala | Amygdala | None | None | None | None |
| P6 | Hippocampus | L | None | None | None | None | Yes | None | Collateral sulcus | Not concordant |
| P7 | Hippocampus | R | Yes | None | Hippocampus; Amygdala and perirhinal cortex | Hippocampus; Amygdala and perirhinal cortex | None | None | None | None |
|  | Amygdala | L | None | None | None | None | None | None | None | None |
| P8 | Hippocampus | R | None | Yes | Hippocampus and thalamus; Hippocampus, thalamus and amygdala | Hippocampus and thalamus | None | Yes | Hippocampus and Thalamus | Hippocampus and thalamus |
| P9 | Hippocampus | L | Yes | None | Hippocampus | Hippocampus | None | None | None | None |
| P10 | Amygdala | L | None | Yes | Hippocampus, thalamus, perirhinal cortex and insula | Hippocampus, thalamus and insula | None | Yes | Hippocampus, amygdala, thalamus, perirhinal cortex and insula; amygdala | Hippocampus, Thalamus and insula |
| P11 | Hippocampus | R | None | Yes | Thalamus and putamen and insula | Thalamus | None | Yes | Thalamus; Thalamus and fusiform gyrus | Thalamus |
| P12 | Hippocampus | L | Yes | Yes | Amygdala and Hippocampus; Thalamus; Thalamus and hippocampus | Hippocampus and thalamus; Thalamus | Yes | None | Amygdala; Amygdala and Hippocampus | Hippocampus |
| P13 | Amygdala | L | None | None | None | None | None | None | None | None |
| P14 | Amygdala | R | None | None | None | None | None | None | None | None |

*Lat* lateralization of the marker, *R* right, *L* left
A summary of results of « SEEG-triggered analysis » and « Continuous-analysis» in each patient for a particular SEEG marker is detailed in the table. We reported if a "mesiotemporal" or a "thalamic" ICA were present either in SEEG triggered or continuous analysis. In the "structure correlated" column we specified the exact structure in the mesiotemporal lobe with which the ICA had a significant correlation (as indicated by the localization of the SEEG contact(s) on patient's MRI) and, finally, the results of source localization when applicable (high goodness of fit)

in all the seven patients (8/9 components). Moreover in 7/9 components the localized source included the exact brain structure with which the component correlated on SEEG (Supplementary Data 1; Supplementary Fig 1).

Source localization showed sensitivity for mesial structure detection of 0.81 and specificity of 0.93 (calculated by using source localization on lateral components—see methods for details).

**ICA, independently of SEEG markers, can detect mesial sources.** To evaluate whether ICA on MEG could detect mesio-temporal sources independently of the information coming from SEEG, we computed ICA on the continuous MEG signals (without SEEG triggering, from now on called "continuous analysis"). We then computed correlation between ICA and SEEG on the periods around the SEEG spikes of the components correlating with mesial structures (with the same procedure as for "SEEG-triggered analysis").

We found significant correlation with mesio-temporal regions alone in six patients (P1, P2, P3, P4, P6, and P12) (Table 1). The

isolated mesio-temporal regions with which the continuous ICA signals correlated were hippocampus (four patients: P1, P2, P3, and P4), amygdala (two patients: P4, P12), amygdala and hippocampus (one patient: P12), collateral sulcus (patient P3, P4, and P6) and perirhinal cortex (P3, P4) (Table 1).

We evaluated the ICA topography of these mesial sources and found a common characteristic: these topographies were generally widespread (Fig. 1b), i.e. the isofield lines are distant, delineating a "distant maxima" dipolar field, corresponding to a deep equivalent dipole[37]. This kind of topography was different from those associated with classical neocortical sources and could help clinicians to identify mesial sources.

Source localization analysis, when applicable (14/20 components with a high goodness of fit—see Methods section), was able to confirm the mesial origin of the components defined independently of SEEG triggers in four patients out of six (9/14 components). In three patients (4/14 components) the component was localized in the exact brain structure correlating with SEEG (Table 1 and Supplementary Data 1; Supplementary

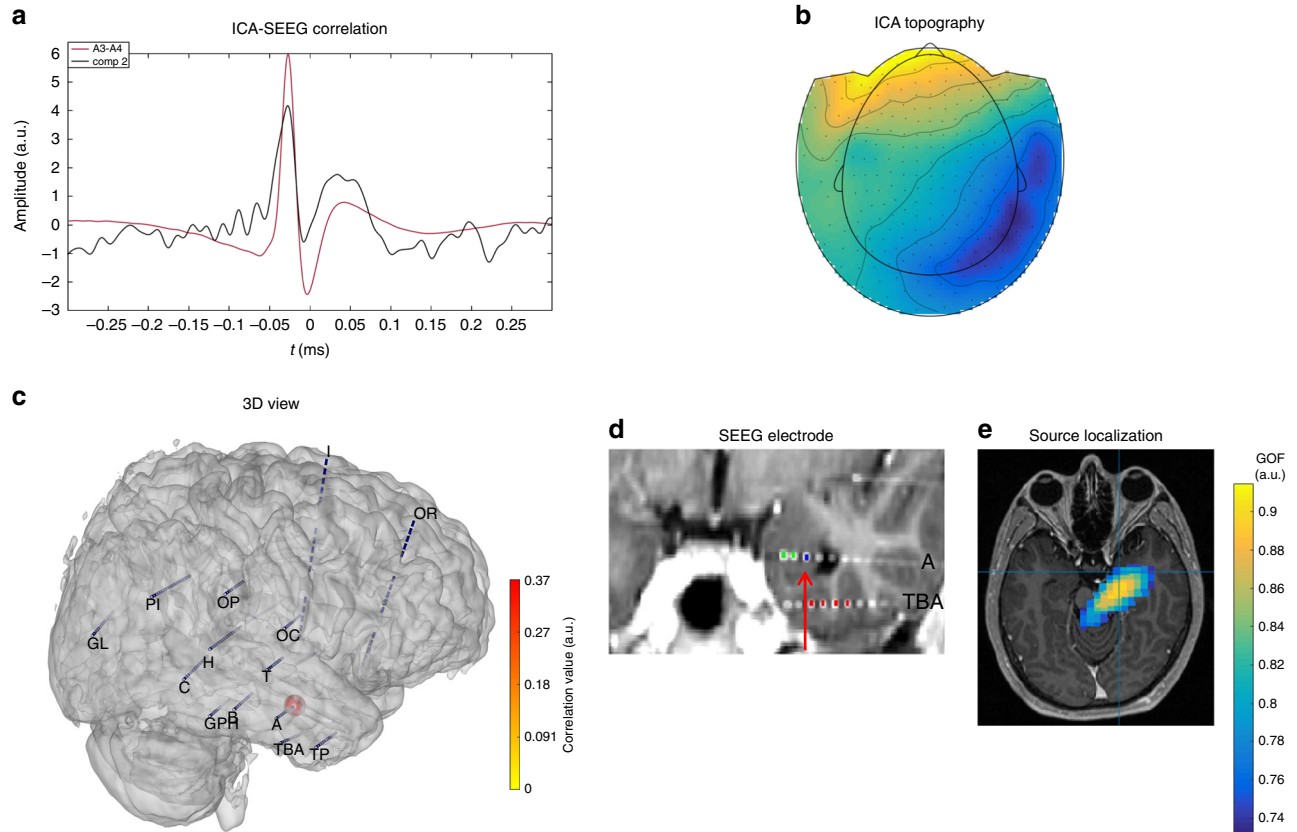

**Fig. 1** Example of purely mesial network, correlated with amygdala in P5. **a** SEEG/ICA correlation: in black the ICA time course and in red the time course on the SEEG contact where significant correlation was observed. **b** MEG topography of the ICA component. **c** 3D representation of MEG-SEEG correlation. Color (from yellow to red, refer to color bar) and sphere dimension correspond to the correlation value. **d** MRI (3D T1) with reconstruction of SEEG electrodes (showing electrodes A and TBA on the right hemisphere): the arrow indicates the contact (the most mesial within the bipolar derivation) with the highest correlation value. This contact, within electrode A, is located in the right amygdala. **e** Source localization (single dipole) of the ICA component overlaid on patient MRI, showing a confidence interval that includes regions sampled by the mesial contacts of electrode A

Figure 1). It should be noted that source localization of deep activities is a difficult issue (Discussion section).

**ICA permitted to disentangle local from extended networks**. In order to describe the intracranial networks as distinguished by ICA, we classified the components according to the structures with which they correlated. We consequently identified mesial (M) networks, restricted to mesial temporal structures (i.e. amygdala or hippocampus), lateral (L) networks involving only lateral neocortical structures, and mesio-lateral (ML) networks involving mesial and lateral structures. We classified as extended limbic (eL) those networks that correlated with structures of the limbic circuitry outside the temporal lobe, such as the thalamus, the insula and the orbitofrontal cortex.

This analysis revealed heterogeneous expression of interictal networks across patients. Some patients showed only one network type, while in other patients, different network types could coexist (e.g. M and ML). Mesial network as the only type was seen in four patients (P1, P2, P6, and P9). A detailed description of network types is reported in Supplementary Data 1 and in the Methods section.

We illustrated P5 as an example of a patient with two network types. For this patient, we could evidence that one ICA component correlated with amygdala alone (Fig. 1) and another ICA component correlated with both mesial and lateral sources (Supplementary Data 1) describing an extended meso-lateral network. Interestingly, topographies are dissimilar between the

two components: the mesio-lateral component showed a topography similar to the one found on MEG sensors during spikes (see below- Methods part) whereas the mesial ICA component had a completely different topography, constituted by a "distant maxima" dipolar field, suggesting a deep source (confirmed by source localization) (Fig. 1b, e).

To evaluate the level of redundancy of the different types of analysis across the same patient (SEEG-triggered vs. continuous analysis or the between the different filtering settings), we performed a hierarchical clustering analysis (see Methods section for details). In summary, we found that the mesial components were rarely clustered together (only in one patient for M components and in 3 patients for meL components) across the different analyses, confirming the importance of several strategies for retrieving mesial activities on MEG.

**Thalamus may produce a signal detectable from surface**. In four patients (P8, P10, P11, and P12), by means of the "SEEG-triggered analysis", we found at least one ICA component correlating with both a mesio-temporal structure and the thalamus (Fig. 2). The same patients, except for P12, showed thalamic correlation also using "continuous analysis". In two patients (P11 —"continuous analysis"–P12—"SEEG-triggered analysis"-) we found the thalamus as the only structure with which the ICA component is correlated with (Fig. 3). Source localization confirmed the correlation analysis (Fig. 3).

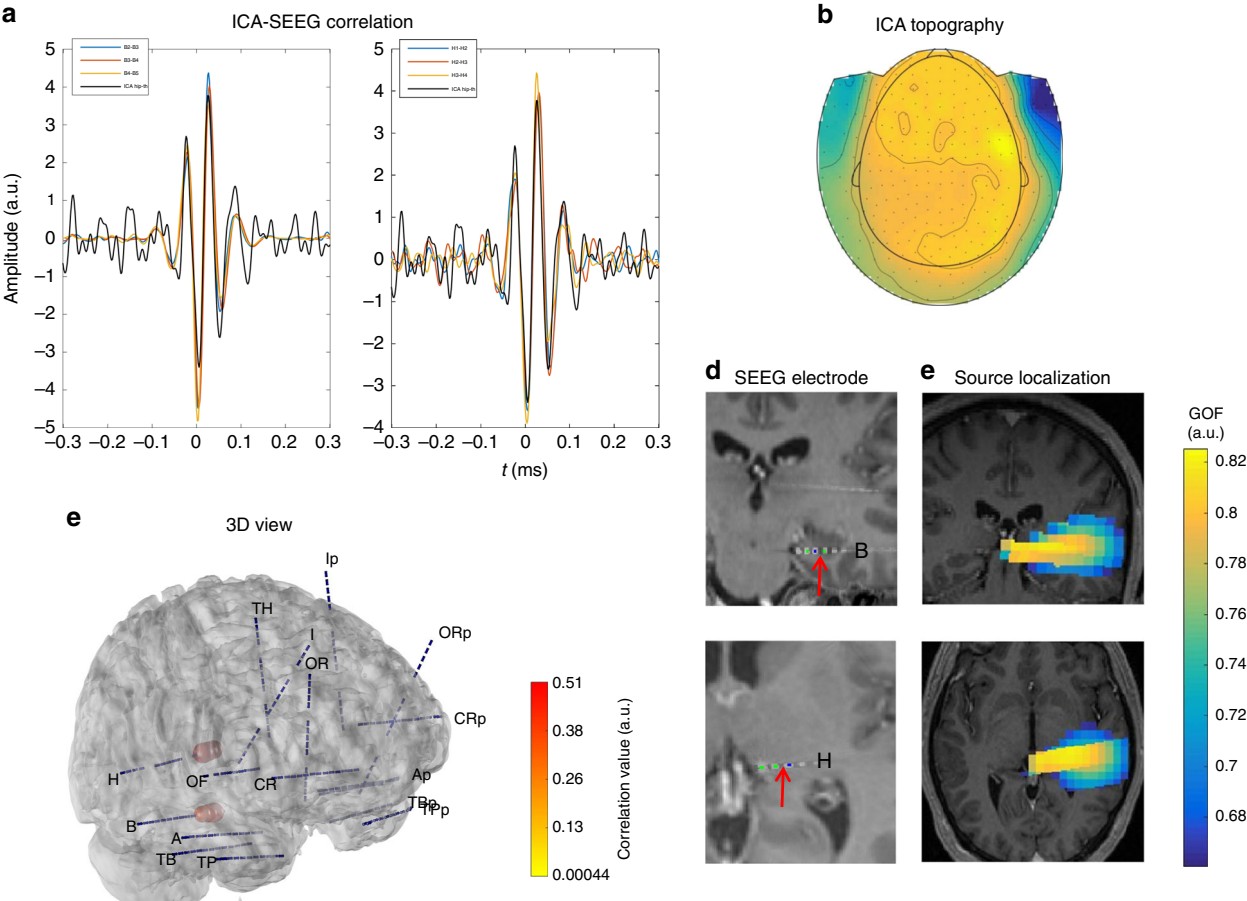

**Fig. 2** Extended limbic network. ICA component correlated with thalamus and hippocampus in P8. **a** In black, ICA time course and in color, time courses on the SEEG contacts where significant correlation with ICA was measured. **b** MEG topography of ICA component. **c** 3D representation of MEG-SEEG correlation. Color (from yellow to red, refer to color bar) and sphere dimension correspond to the correlation value. **d** MRI (T1) with reconstructed SEEG electrodes; the arrow indicates the contact with the highest correlation value, located in the right hippocampus (B) and right thalamus (H). **e** Source localization (single dipole) of the ICA component overlaid on patient MRI, which includes regions sampled by mesial contacts of electrodes B and H

Moreover, we also found ICA components which correlated, together with other brain regions (including lateral neocortical regions), with some other deep extra-temporal structures (Supplementary Data 1): this was the case for the insula in three patients (P10, P11, and P13), putamen (P11), and the orbitofrontal cortex in two patients (P12, P13).

**The signal to noise ratio of ICA components is generally low**. We aimed to assess a value of reference for visibility at a MEG sensor level of the different brain structures, notably of amygdala and hippocampus. Each ICA component, in fact, could be seen as the surface representation of the brain structure/s (SEEG signals) with which it correlated. Given this assumption, the signal to noise ratio (SNR) of the ICA components would represent such a value of reference. These results could be eventually further generalizable and applied to other neuroscientific domains, such as cognition studies.

We thus computed the SNR of each ICA component back-projected on the MEG sensors. We measured the SNR in relation to the background noise and in relation to the MEG data at the moment of the event of interest (100 ms around the SEEG spike), and we repeated these measurements for an increasing number of averaged events.

We found that 45 out of 115 ICA components were visible with respect to the background and only 5 out of 115 were visible at the moment of the events (Supplementary Data 1). This means that in most cases ICA was able to identify signals otherwise hidden by other simultaneous signals. By averaging an increasing number of events, we did not always find progressive augmentation of visibility. An example of augmented visibility with an increase of the averaged events is illustrated in Fig. 4 for a mesial network correlating with hippocampus.

To verify whether the visibility of the ICA components was influenced by study variables (number of markers, brain structure marked, network type, filtering, triggered analysis, lateralization, number of structures implicated in the network) we performed statistical analysis (based on a generalized logistic mixed model). We found that filtering influenced visibility in relation to the background noise (ICA calculated with a 2–60 Hz filter is 2.9 more probable to be visible than ICA calculated with 12–60 Hz filter, odds ratio CI: [1.16–7.14]). The brain structure marked also influenced visibility: ICA was 6.5 times more likely to be visible when calculated using hippocampus markers compared to amygdala markers. However, in this case, a great heterogeneity was found (odds ratio CI: [1.28–33.04]). No significance was found for the number of markers, "SEEG-triggered analysis" vs "continuous analysis", and number of structures involved in the network. A significant effect of the network type and the lateralization was also found, but the level of increased probability was small or too heterogeneous.

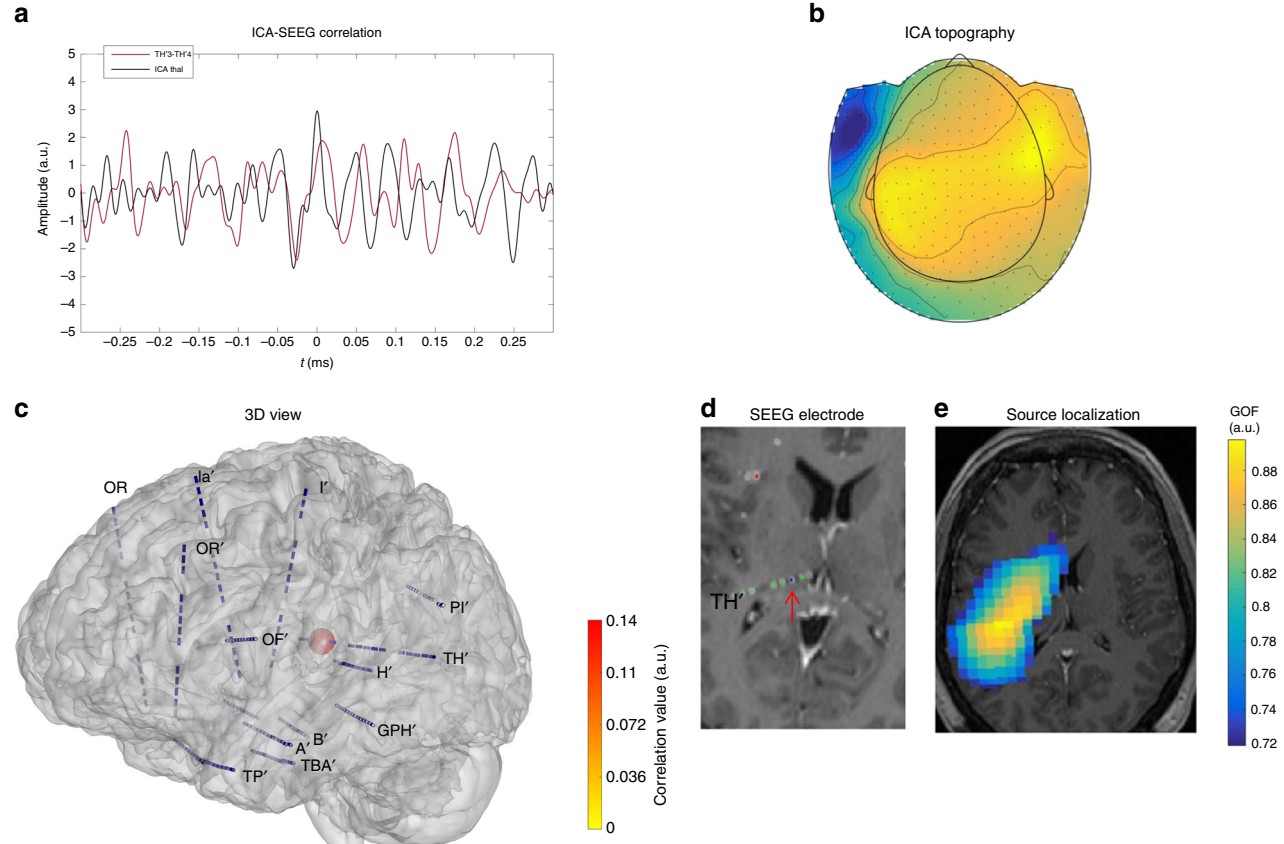

**Fig. 3** ICA component correlated with the thalamus in P12 ("SEEG-triggered analysis"). **a** In black, ICA time course and in color (red), time course on the SEEG contact where significant correlation with ICA was measured. **b** MEG topography of ICA component. **c** 3D representation of MEG-SEEG correlation. Color (from yellow to red, refer to color bar) and sphere dimension correspond to the correlation value. **d** MRI (T1) with reconstructed SEEG electrodes (showing electrode TH'); the arrow indicates the contact with the highest correlation value, located in the left thalamus. **e** Source localization (single dipole) of the ICA component overlaid on patient MRI, which includes regions sampled by mesial contacts of electrode TH'

## Discussion

We report the possibility of recording activities generated in specific mesial temporal lobe structures including the amygdala and the hippocampus using MEG, in patients with focal drug-resistant epilepsy. Our results are based on simultaneous MEG and SEEG recordings, which allow assessments of visibility on exactly the same signals. Importantly, we used independent component analysis (ICA) as a way to disentangle networks in which activities were mixed at the sensor level. We also obtained evidence of MEG detectability of other deep brain structures outside the temporal lobe, such as the thalamus.

Hippocampal activity has long been thought to be undetectable from the surface due to its "closed field" configuration[4]. In short, because of the convoluted shape of the hippocampus, its activation was considered to produce no magnetic field because of the mutual cancellation of oppositely oriented generators. However it has been elegantly modeled[9,38] that there might not be as much cancelation as expected, resulting in a possible detection by MEG[9]. Our results suggested that this is indeed the case, at least for activities such as epileptiform discharges, and for magnetometers which are potentially more sensitive to deep sources than gradiometers.

Based on prior assumption on the network organization of the spiking activity in temporal lobe epilepsy on SEEG study[19], we used ICA with the aim of separating the different components of the epileptic network that could occur together or independently[29]. Correlation between ICA and SEEG channels was studied at zero lag and using several complementary methods. We

showed, with this method, that it is possible to detect and differentiate from the surface recordings the different components of the epileptic network involving spiking activity in hippocampus and amygdala. Heterogeneous and diverse networks behaviors were observed among all patients: different patterns of co-activated brain structures as recorded by intracranial electrodes were extracted from the surface. Some networks evidenced by ICA were characterized by correlation with few mesial structures (Fig. 1), while others, more extensive, reflected the involvement of multiple brain structures within the network[39] (Supplementary Data 1). Indeed, it has been demonstrated on neuroimaging studies[40,41] and by intracranial recordings[3,19,42] that brain regions distant from the mesial temporal lobe are often involved in epileptogenic limbic circuitry, together with hippocampus, amygdala, and neocortical mesial structures[43].

Previous studies explored the visibility of mesial structures on surface using simultaneous intracranial and surface recordings. Such recordings are the only way to ensure that exactly the same activity is measured in depth and at the surface, allowing moreover to use single-trial analysis as a source of information on the relationships between signals[35]. Dalal et al.[44], in a SEEG-MEG study, used zero-lag correlation to find the MEG sensor that best correlated with the hippocampus traces considering the zero lag a condition sufficient to eliminate spurious correlation with other brain regions. In our study, however, we evidenced that even at zero lag the limbic network could be already extended to other lateral sources; it is therefore preferable to explore correlation

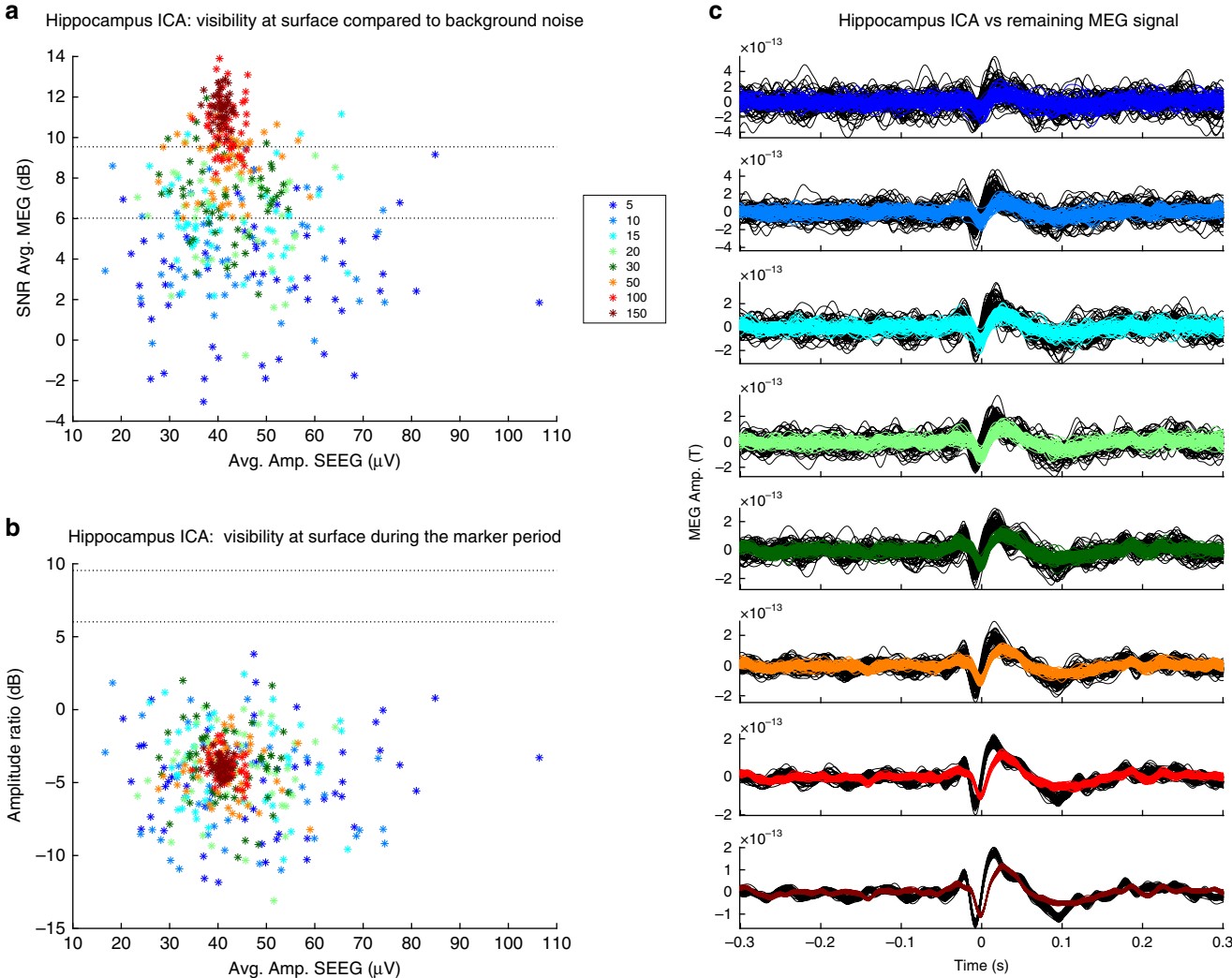

**Fig. 4** SNR evaluation of a ICA component correlated with hippocampus ("Hippocampus ICA") in the "continuous analysis". **a** ICA visibility compared to background noise: the stars indicate the number of markers to average to obtain a certain visibility (SNR). **b** ICA visibility during the marked period. **c** Average time course of the ICA component in relationship to the MEG signal on the sensor where the ICA topography has the maximal absolute value. Each color line refers to the number of events that had been averaged (refers to figure legend)

with all the SEEG channels at the same time to verify the actual epileptic network extension.

Koessler et al.[32] demonstrated on simultaneous SEEG-EEG recordings that pure intracranial mesial networks were visible on surface EEG, after averaging a large number of spikes (48–440) recorded on SEEG[21]. They used a classification of spikes, involving only hippocampus or a mixture of hippocampal and neo-cortical activity. Our study on SEEG-MEG went beyond this result, demonstrating that ICA could automatically separate the different mesial temporal activities, also including pure hippo-campal activity.

An important limitation of our study comes from the spatial sampling of SEEG, which is only partial in any given patient. It is thus possible in principle that MEG-ICA captures a source not sampled by SEEG, hampering the correlation analysis. None-theless, it is important to note that SEEG electrodes sample both mesial and lateral aspects of the temporal cortex and that clinical implantation aims at covering the hypothesized epileptic network (including the epileptogenic zone and the propagation network). This usually results in a reasonably high sampling of the different brain areas supposed to be involved in the ictal and interictal activities. In most patients, implantation was bilateral, the

maximum number of electrodes for a single patient reaching 19, with up to 15 contacts per electrode.

In order to reduce the risk of capturing a source not visible by SEEG, we applied a series of strict tests to the correlation analysis, including zero-lag correlation measures across both temporal and trial dimensions. Source localization was applied to the ICA topographies to evaluate if it was possible to retrieve mesial components from surface analysis. The mesial nature, and even more precisely the exact structure evidenced with SEEG corre-lation analysis was confirmed by source localization in the majority of cases, especially for the "SEEG-triggered" analysis that reveals to be more accurate than the "continuous analysis".

Regarding the mesial components not confirmed by source localization, we hypothesize that current methods of source localization are not fully adapted for the localization of mesial structures. The depth of the generators, their geometry, the low signal to noise ratio and the difficulty in obtaining noise covar-iance matrix for ICA topographies all hamper the current tech-niques. Future studies should be done to optimize source localization methodology to improve detection of mesial struc-tures. We believe that in our cohort, given the large SEEG implantation and the careful multistep correlation study between

ICA on MEG and SEEG channels, it is more reliable to identify the origin of the surface components based on multistep MEG ICA-SEEG correlation analysis than on current models of source localization. Despite all the limitations, an overall sensitivity of 0.81 for source localization of mesial ICA components was found.

Interestingly, we have shown that is possible to retrieve a pure mesial network from continuous signals, i.e. without information coming from intracranial recordings ("continuous analysis" Table 1). Surprisingly, in two cases (P4 and P6) "continuous analysis" performed better for mesial network detection than "SEEG-triggered analysis" (Table 1). This could be explained by the fact that hippocampus could eventually leave a signature on the surface outside the epileptic spike events. It is for instance possible that specific oscillatory patterns generated in the mesial structures could be detected on the surface. In temporal lobe epilepsy, alteration of resting state networks outside the interictal events has already been reported on high density EEG data[45] and on SEEG[46]. We obtained better results in the SEEG-triggered analysis than in the continuous analysis, which suggests special care has to be taken in analysis without SEEG. Another point of discussion is the fact that multiple ICA analyses were performed on the same patient data, possibly generating redundant results. However, when focusing on mesial ICA components we verified with the clustering analysis that few were similar.

One important finding of our work was the participation in the epileptic network together with hippocampus and amygdala of subcortical structures (such as thalamus and putamen) and of the limbic cortex outside the temporal lobe (orbitofrontal cortex and insula) (Supplementary Data 1). Our results suggested the detectability of such deep brain regions on MEG as previously modeled[9,47] and reported[48,49]. It has long been argued that focal epilepsy is a cortico-subcortical disease[50,51] and the role of thalamus in epilepsy has gained renewed interest[52–55]. Taken together, these data offer the opportunity in the future to non-invasively study the actual extension of the limbic epileptic network to such brain structures believed to be undetectable on surface. Furthermore, thalamus detectability as a part of the epileptogenic network could potentially represent an important factor for epilepsy surgery prognosis[53,56].

We also aimed to evaluate the visibility of the different ICA components by calculating their relative amplitude on surface measurements (SNR). In particular, for mesial networks, the SNR could be taken as a reference to be applied to other domains, i.e. cognitive studies. We showed that in most of the cases, either triggered by SEEG markers or not, and even after averaging a large number of events, SNR values were too small to be seen without ICA as they were hidden by other simultaneous signals. The ICA was thus confirmed to be essential to the correct detection of the different interictal networks.

We noticed that averaging a few markers could be sufficient in certain patients to detect amygdalar activity at the surface (P4; P5), while for other patients a larger number of spikes marked on amygdala (P7; P13; P14) was not sufficient to reach detectability (Supplementary Data 1). Still, in the majority of cases, adding events to the averaging resulted in increased visibility (Fig. 4). The most important factor determining visibility of ICA networks was filtering, with a higher visibility of ICA component when using a larger band filter (2–60 Hz vs 12–60 Hz). Also, the structures marked on SEEG influence the visibility, the hippocampus being more visible than the amygdala, but with a larger variability of effects. Thus, there seem to be factors other than number of spikes, structure, and extension of the network that influence visibility. Possibly some variability arises from the state of the patient at the moment of recording (awake, light sleep). It is also possible, as suggested by the better detectability of mesial structure with "continuous analysis" compared to the "SEEG-triggered" one in two patients, that markers other than epileptic spikes, should be considered to study mesial temporal structure visibility. Moreover, orientation of the dipole was not considered, which is an important parameter arduously evaluated from SEEG.

The use of ICA to automatically separate sources from surface signals has multiple advantages in clinical practice: it is rapid and provides a topography which is unique for the network of interest. Further studies are needed to validate this technique on a larger number of patients for routine practice. Finally, we recommended evaluating the time courses of the ICA components which present this particular topography characterized by "distant maxima" dipolar fields that could help in the identification of components correlated with mesial sources. This kind of "large" topography, characterized by a broad distance between isofield lines, has already been described on EEG[37] and more recently in an EEG-fMRI study[57]. We could confirm, thanks to the simultaneous SEEG recording and to source localization, that this kind of topography could indeed reflect a deep generator.

Detecting activities non-invasively from deep brain structures is of broad interest in both basic and clinical neuroscience, as it could potentially tremendously improve our understanding of the limbic system circuitry in humans. Thus, it may have implication for physiological processing as memory/emotion and for aberrant activity as in neurological or psychiatric disorders.

## Methods

**Simultaneous SEEG-MEG recordings.** We analyzed simultaneous recordings of 14 patients undergoing an intracerebral stereotaxic EEG (SEEG) investigation for pre-surgical evaluation of focal drug-resistant epilepsies at the Epileptology and Cerebral Rhythmology Unit, APHM, Marseille, France. The methodology for the setup of simultaneous SEEG-MEG recording is detailed in Dubarry et al.[35] and technical specificities, with regards to difficulties inherent to simultaneous recording are presented in Badier et al.[36]. In summary, we recorded from 10 to 30 min of simultaneous SEEG and MEG during a resting state period (patient with eyes closed, relaxed, possibly sleeping). Signals were acquired on a 4D Neuroimaging™ 3600 whole head system at a sampling rate of 2034.51 Hz. The simultaneous SEEG-MEG recording was carried out following at the end of the long-term video-SEEG monitoring period. A total of 248 magnetometers for each patient with noise subtraction from reference sensors were recorded. Simultaneously we also acquired a range of 70–249 SEEG contacts[58] (total contacts recorded: 2383 mean for patient 167, SD ±61) as well as EOG and ECG channels. Intracerebral EEG electrodes were implanted in stereotactic conditions[59] either orthogonally or obliquely. The trajectory of each lead was based on the individual anatomy and vascular constraints of the patient. SEEG implantation was decided for each patient according to the hypothesis about the epileptogenic zone derived from clinical, neurophysiological and imaging data available during non-invasive pre-surgical evaluation[60]. The electrodes had a diameter of 0.8 mm, and contained 10–18 contacts, each 2 mm long and separated from each other by 1.5 mm (Alcis, Besançon, France). SEEG signals were sampled at 2048/2500 Hz and recorded on a hard disk (16 bits/ sample) using no digital filter. A hardware high-pass filter was set at 0.16 Hz at −3 dB. Because a separate system was used to record the SEEG signal for some patients (patients from 3 to 14), regular triggers with time jitters (inter-trigger range 3000–3500 ms) were sent to both SEEG and MEG systems. Signals were co-registered offline based on these triggers, with in-house code written in Matlab (Mathworks, Naticks, MA).

**Patients and records selection.** The only criterion for patient selection was the presence of at least one electrode located in the hippocampus head or in the amygdala showing interictal epileptic activity ("interictal spikes"). Fourteen patients were studied (8 female). Patients' mean age at recordings was 31.5 years, mean age at epilepsy onset was 18.2 years and mean epilepsy duration was 15 years. Clinical information, including epilepsy characterization, neurophysiological recording, and neuroimaging data, were collected and analyzed (Table 2 for a summary). The patients signed a written consent for simultaneous recordings. This research has been approved by the relevant Institutional Review Board (Comité de Protection des Personnes, Sud-Méditerranée I, ID-RCB 2012-A00644–39).

Recordings disturbed by severe SEEG or MEG artifacts or technical problems were excluded from this study. For each patient CT-scan/MRI data fusion was performed to accurately check the anatomical location of each SEEG contact along the electrode trajectory according to previously described procedures[61]. For this purpose, we used our in-house software GARDEL (a Graphical User Interface for Automatic Registration and Depth Electrodes Localization). GARDEL is a Matlab-based tool which co-registers MRI to the CT-scan, automatically segments and precisely localizes contacts of depth electrodes by image processing[62].

**SEEG pre—processing and marking**. We choose a bipolar SEEG montage for the analysis. A bipolar montage is composed by channels constituted by the difference of adjacent electrodes. We only included channels with at least one electrode located in the gray matter. A total of 1016 bipolar channels was studied (range 39–129, mean 73 SD 26). Bipolar montage was preferred to a referential montage (common reference), because it is less sensitive to volume conduction thus allowing a more precise spatial localization of the transient events. It also removes reference effects that could affect correlation measures. For each patient, at least one SEEG channel inside hippocampus or amygdala was selected for visual spike marking (the 2 contacts of the bipolar channel selected being both inside the same structure). We marked hippocampus as first choice and if no electrode was present in the hippocampus head or it was at his border and an electrode was present in the amygdala we selected the amygdala electrode for spike marking. When a patient had a bilateral implantation and spikes were present bilaterally both sides were studied (P3; P7). All signals were reviewed using our in-house software Anywave[63] (available at http://meg.univ-amu.fr/wiki/AnyWave). Interictal epileptiform spikes were marked on the selected channels. For an optimal recognition of the transient event we applied to SEEG recordings a high-pass filter at 0.03 Hz and we used two windows simultaneously 1) for spike identification (gain 500 μV cm$^{-1}$, time scale 0.5 s cm$^{-1}$) 2) for spike marking (gain 200 μV cm$^{-1}$, timescale 0.2 s cm$^{-1}$). We positioned the instantaneous SEEG mark on the peak of the maximal positive or negative deflection of the spike (Fig. 5a1).

**MEG processing and ICA analysis**. Data analysis on MEG was performed using Anywave software and Matlab scripts (The Mathworks Inc., MA, USA). All relevant data and computer code are available from the authors. All MEG channels were reviewed and channels with artifacts or flat signal were removed from the analysis. Two ICA decompositions were compared, performed either: i) on the MEG traces segmented around the SEEG triggers (Fig. 5a2, a4) or ii) on the continuous MEG signals without information from SEEG. For each ICA analysis, we used two different band filters, one in the 2–60 Hz band and one in the 12–60 Hz band. We decided to test 2 distinct filters as the results of processing epileptiform discharges can be influenced by low frequencies[64]. MEG signals were band pass filtered to 1–170 Hz (FIR filter type) and downsampled at 512 Hz in order to facilitate data analysis. A time window of 0.6 s, centered on each SEEG spike trigger was selected for MEG data segmentation for the analysis (part i) above). The continuous analysis (part ii) above) did not rely on segmentation around markers. A total of 20 ICA components were extracted for each analysis.

**Depth-surface temporal correlation: ICA component selection**. In order to ensure a strong link between SEEG and MEG signals, we performed two complementary correlation analyses followed by visual inspection. We firstly computed the correlation (corrcoeff Matlab function) of single-trial ICA time courses with all events concatenated along the time dimension (Fig. 5ba). In order to render the measures more Gaussian, a Fischer transformation was applied to the correlation coefficient as follows:

$$\eta = \frac{1}{2} \times \log\left(\frac{1+r}{1-r}\right),  \quad (1)$$

with $r$ the Pearson correlation and $\eta$ the transformed correlation coefficient. Statistical analysis (local false discovery rate, lFDR[65], with a threshold set at 0.2 as proposed in ref. [65]) was applied to evaluate which SEEG contacts were significantly correlated with the ICA component. The local FDR is an empirical Bayesian technique that assumes that the data histogram is a mixture of a Gaussian noise (representing the central part of the histogram) and an unknown distribution (signal of interest in the tails of the histogram). The lFDR assesses whether a given values 'stands out' of the noise. It is a "local" measure, quantifying the threshold value at which the ratio between the estimated false detection and the total detections fall below 0.2, and not an integral between then the threshold and infinity as in classical FDR. Thus, the 0.2 value can be considered as an equivalent of the $p$ or $q = 0.05$ in other corrections for multiple comparisons.

ICA components with at least one significant SEEG correlation were selected. To ensure a strong correlation between selected component and SEEG contacts throughout the different trials, we further studied the inter-trial correlation (ITCOR)[35] at each time point along the trial dimension. In other words, we did not only ensure that the overall shape of signals was correlated at zero lag, but also that signal amplitudes were fluctuating jointly across trials. We applied subsequently the same statistical analysis as in the previous step (Fig. 5bb).

Only ICA components with temporal correlation confirmed by ITCOR analysis were retained. Retained ICA components' topographies and time courses were visually reviewed to ensure that correlation was at zero-lag and to check for their non artifactual nature (i.e. blinking or cardiac artifacts). We visually checked the average time courses (SEEG/ICA) and we ensured that the visible correlated activity on the average time course corresponded to the same period of significant correlation in the ITCOR analysis as visualized in the box (Fig. 5bb, c). Only components with temporal concordance between time course and ITCOR were retained. Moreover, all selected ICA were screened by one reviewer (FP) and then reviewed for confirmation by two expert reviewers (CGB and JMB). After this multistep study, we were confident of the strong link between the selected ICA components and SEEG.

Temporal correlation analysis showed 262 components. After the full selection process 115 ICA components were finally retained.

**ICA component classification**. The selected ICA components were classified according to the anatomical location of SEEG contacts presenting significant correlations. Based on Bartolomei et al.[19] (with slight modifications), we defined

**Table 2 Patient characteristics**

| Patients | Age | Sex | Epilepsy type | Epilepsy onset (years) | Epilepsy duration | Implantation | SOZ | MRI |
|---|---|---|---|---|---|---|---|---|
| P1 | 25 | F | L Fr-T | 7 | 18 | Fr-T, bilat L > R | R Fr-T | Negative |
| P2 | 34 | M | R mesio T | 9 months | 33 | T-perisylvian orbitoFr, bilat R > L | R mesio T | R HS |
| P3 | 46 | F | L mesio T | 28 | 18 | T-perisylvian orbitoFr, bilat L > R | L mesio T | L HS |
| P4 | 29 | F | Bi-T | 20 | 9 | Bilat T P O, R > L | Bi T | L T-O heterotopic lesion and L hipp dysgenesis |
| P5 | 21 | M | R latero T | 15 | 6 | T-perisylvian orbitoFr and P, bilat R > L | R lateral T | Negative |
| P6 | 19 | M | L Orbito-Fr | 2 | 17 | Fr-T and P, bilat L > R | L orbito-Fr | Negative |
| P7 | 34 | F | R mesio T | 32 | 2 | Fr- T and P, bilat L > R | R mesio T | Negative |
| P8 | 38 | F | R mesio T | 19 | 19 | Fr-T and P, bilat | R mesio T | Bilat amygdalo-hippocampus hypersignal |
| P9 | 37 | F | L Mesio T | 38 | 9 | Fr-T and P, bilat L > R | L mesio T | Augmented L amygdala volume |
| P10 | 26 | M | LT plus | 5 | 21 | Fr-T and P bilat L > R | L mesio T | L hip and para-hipp hypersignal, asym hippocampi |
| P11 | 36 | M | R T cavernoma operated | 23 | 10 | T posterior and Fr, bilat R > L | R latero T + orbitoFr | R T lobectomy for cavernoma |
| P12 | 42 | F | LT plus | 24 | 18 | Fr-T and P, bilat L > R | L mesio T | Negative (hypertophic amygdalae) |
| P13 | 33 | M | Bi T | 11 | 22 | Fr-T bilat | Bi T | Negative |
| P14 | 21 | F | R latero T | 13 | 8 | T- insular, bilat R > L | R lateral T | Negative |

F: female, M: male, R: right, L: left, T: temporal, Fr: frontal, P: parietal, O: occipital, HS: hippocampal sclerosis, bilat: bilateral

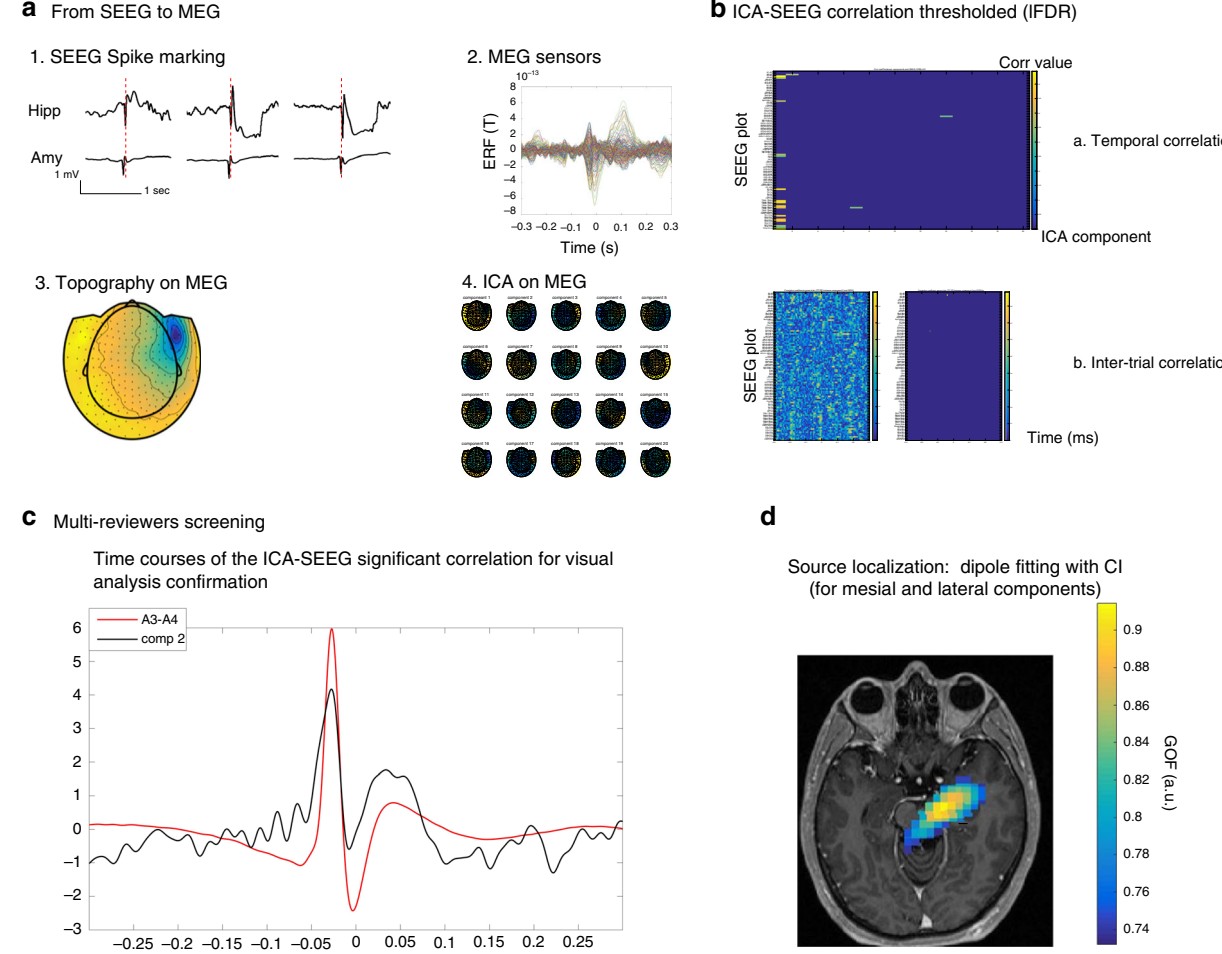

**Fig. 5** Methods. Example of SEEG-triggered analysis in P5. **a** 1. Spike marking on SEEG. Hippocampus and amygdala activity (almost simultaneous): in red marker position based on hippocampus activity 2. MEG sensors time courses at the moment of SEEG markings 3. MEG topography at the moment of SEEG markings 4. ICA topographies (20 components) calculated at the moment of SEEG markings **b**. a ICA-SEEG temporal correlation thresholded (IFDR): on the y-axis SEEG channels, on X-axis ICA components. b. ICA-SEEG inter-trial correlation not-thresholded (on the left) and thresholded (IFDR, on the right): on the y-axis SEEG channels, on X-axis time. **c** Multi-reviewers screening: visual analysis of ICA significant correlated SEEG plot (both in temporal and inter-trial correlation) and ICA time courses and validation of the findings. **d** Source localization applied for mesial and lateral sources

the different ICA network type as:

- Mesial-M—(including temporal lobe mesial structures: hippocampus H, amygdala Amy, internal temporal pole iTP, entorhinal cortex EC, para-hippocampal gyrus, PH, collateral sulcus, SC, fusiform gyrus FusG)
- Lateral-L—(including the external part of the temporal pole, eTP; the occipito-temporal sulcus, OT; the lateral temporo or frontal neocortex, LT, LF;)
- Mesio-lateral-ML—(including the mesial and the lateral structures)
- Extended limbic-eL—(including the temporal structures and/or other extra-temporal limbic structures as the orbitofrontal cortex, insula, putamen or the thalamus. Among the eL we identified networks limited to mesial regions (meL) (Supplementary Data 1).

We could finally describe 31 mesial components (11 SEEG-triggered), 30 mesio-lateral components (16 SEEG-triggered), 31 lateral components (16 SEEG-triggered) and 23 extended limbic components (12 SEEG-triggered) (10 mesial eL – meL, 6 SEEG-triggered).

We questioned whether across the 115-independent components selected, some components were repeated, as we used the same MEG-SEEG data for 4 different analyses in each patient (2 different filtering setting and 2 analyses on triggered and continuous data). To answer to this question, we applied a hierarchical clustering analysis based on correlation across ICA topographies and we defined a similarity index (SI), $n$ clusters/$n$ components. We found that the 115 selected components could be grouped in 63 clusters, SI = 0.54. Interestingly the SI for the only M components was 0.87, confirming the importance of running all the different types of analysis to maximize the chances of detecting hippocampus and amygdala activities. For the other component types, the SI was respectively: for ML 0.6; for meL 0.6; for eL 0.53; and for L 0.58. In 9 patients we indeed found in the same cluster similar ICA components, but retrieved from a different analysis: this was the case for SEEG triggered/continuous analysis in 8 patients and for 2–60/12–60 Hz

filtering in 7 patients. However, this occurred for the M ICA only in P3 (SEEG-triggered and continuous) and for meL ICA in 3 patients: P8 (SEEG-triggered and continuous), P10 (SEEG-triggered and continuous and filtering), and P12 (filtering).

**Source localization on ICA components**. We localized the mesial ICA components (defined as M and meL) in order to evaluate whether deep structures can be retrieved using surface data only. The sensitivity of the method was then calculated.

To verify the specificity of source localization for mesial ICA components, we also calculated source localization of the components classified as lateral (L) and we confirmed their lateral localization.

For the MEG forward calculation, we used a semi-realistic head shape model as implemented in FieldTrip (http://fieldtrip.fcdonders.nl/). Each point of a grid within the brain volume was associated to a triplet of orthogonal dipole. We performed a linear regression to compare each ICA map to the model composed by those triplets and retained the resulting goodness of fit (GOF). A confidence interval was estimated by including all grid points with

$$\mathrm{GOF} > \max(\mathrm{GOF}) - (1 - \max(\mathrm{GOF})). \qquad (2)$$

In other words, we considered the distance between the maximum GOF and 1 as a measured of the noise present in the data. A minimum GOF value of >0.75 was chosen to consider the localization as valid. Sources with GOF < 0.75 were considered as inconclusive for single dipole fit.

For the components showing a tripolar or quadripolar topography (suggesting two dipolar sources) we performed a double dipole fit localization. We thus performed a regression of all possible dipole pairs on the grid and retained only those following condition (2) and belonging to the 5 greater percentile of the F test comparing the fit of two dipoles to the fit of one dipole. The F test permitted to

| Table 3 Specificity of mesial localization | | | |
| --- | --- | --- | --- |
| | Mesial localization (+) (structure included in CI) | Lateral localization (−) (structure included in CI) | |
| Mesial ICA (+) | 27 (21) | 2 | [29] |
| Lateral ICA (−) | 6 | 28 (21) | [34] |
| | 33 | 30 | [62] |

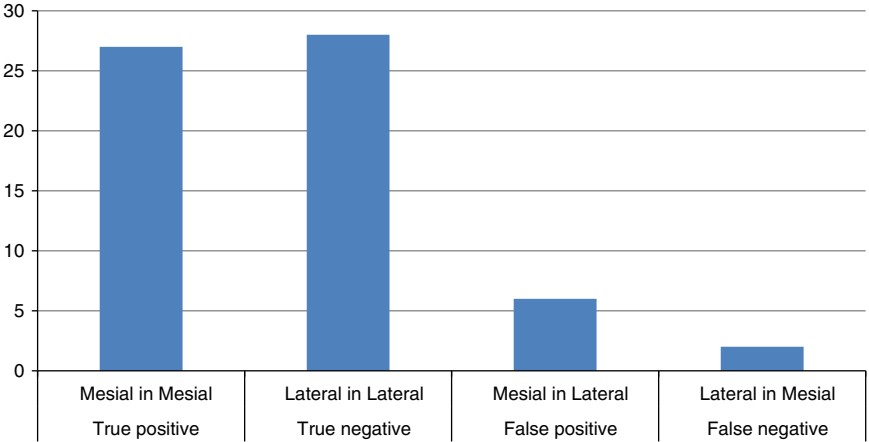

**Fig. 6** True and false positive for mesial ICA localization

rejects pairs of dipoles where only one dipole was significantly contributing to the fit[66]. We obtained a map of the score (sum of pair of dipole respecting these constraints) by grid points. For the double dipole fit localization we choose a GOF > 0.8 to consider them as valid.

In most of cases we used a single dipole fit and kept all localizations within a confidence interval (CI). For seven M topographies (Supplementary Data 1: P1, P3, P7, and P12) and 8 L topographies (Supplementary Data 1: P10, P13, P14) showing at visual analysis a tri- or quadri-polar map, we performed a two-dipole scan[66,67]. In these cases, locations corresponding to "ghost sources" (for example in the middle of two plausible sources) were not considered.

We analyzed the ICA dipole maps on the patients' MRI and we checked visually:

1. If source localization was in the mesial regions of the brain for M and meL components or in the lateral regions for L components.
2. If the SEEG structure with the highest correlation with the ICA component was included in the confidence interval of the dipole scan.

We estimated a sufficiently large CI because of the localization errors reported in the literature to estimate mesial sources[68]. We reported in Supplementary Data 1 whether the maximal value of source localization was in mesial or lateral brain structures or in other structures and if the structure correlating with SEEG was included in the CI.

Regarding mesial components, all the 10 *me*L and 17 among 31 M components were localized in mesial regions using source localization. Moreover, in all the *me*L components and in 11 M components, the brain structure with the highest correlation value was included in the confidence interval of the localization map. Eight M components (6 in the continuous analysis) could not be localized with confidence (GOF < 0.75 for single dipole fit or GOF < 0.8 for double dipole fit) and six M components were not concordant with source localization.

Regarding lateral components, we found 28 out of 31 components localized in the lateral regions, 1 ICA component was not localizable due to the low GOF (SEEG-triggered analysis) and 2 components (continuous analysis) were localized in the mesial structures. In 21/28 cases the brain structure with the highest correlation value was included in the confidence interval of the localization map (Table 3, Fig. 6).

In summary, sensitivity for mesial components was 0.81 and specificity 0.93. The sensitivity for specific brain regions (when the SEEG structure with the highest correlation with the ICA component was included in the confidence interval of the dipole scan) was 0.77 for mesial ICA and 0.75 for lateral ICA.

**Computation of SNR for component of interest.** We questioned the visibility of each networks found by ICA:

i. Is the network detectable at a MEG sensor level?
ii. Can the network be differentiated from the co-occurring signal?
iii. Is the detectability of the different networks improved by averaging an increasing number of markers?

To answer these questions, we computed the signal to noise ratio (SNR) of the MEG reconstructed ICA components (ICA components were back projected to the MEG sensor level). We considered as visible a signal with an SNR of at least 10 dB (corresponding approximately to an amplitude ratio of 3 between peak and background noise). The MEG channel where the ICA topography had the maximal absolute value was selected for further analysis. We also reconstructed the background noise by rejecting the selected ICA component from the MEG signal. The following measures were performed.

i. To evaluate if the ICA components corresponding to the different networks (M, ML, eL, L) were visible on the MEG sensors we measured the SNR value comparing the highest amplitude of the projected ICA component (channel with maximum amplitude) to the background noise (calculated on a baseline time window outside the event). The SNR was thus defined as:

$$\mathrm{SNR} = 20\log_{10}(\max(|\bar{s}|)/E\{\tilde{\sigma}\}), \quad (3)$$

$\bar{s}$ is the average across the SEEG marked spikes of the reconstructed signal of interest on the selected MEG sensor in a small window $[-0.05, 0.05]$ centered on the spikes; $\tilde{\sigma}$ is the standard deviation of the background activity outside the spikes $[-0.3–0.05] \cup [0.05, 0.3]$; $E\{.\}$ is the expectation value (mean) across spike markings.

ii. To test whether the different networks were visible in relation to concomitant activities, we calculated the amplitude ratio of the highest amplitude of the back-projected ICA component over the highest amplitude of the background (sum of all other components), at the time of the event (considered ± 0.05 s around the instantaneous SEEG marker). The SNR was thus defined as:

$$\mathrm{SNR} = 20\log_{10}(\max(|\bar{s}|)/\max(|\bar{s}_c|)) \quad (4)$$

$\bar{s}$ is the average across the SEEG marked spikes of the reconstructed signal of interest on the selected MEG sensor in a small window $[-0.05, 0.05]$ centered on the spikes; $\bar{s}_c$ is the average across the SEEG marked spikes of the reconstructed background (all other ICA components) on the same MEG sensor in the same (0.1 s) time window.

iii. These SNR measures were computed for an average performed on an increasing number of events (e.g. 5; 10; 15; 20…) randomly drawn from all the spikes in the same channel. This procedure was repeated 50 times. We finally identified the minimum number of trials to average to have 75% of visibility above 10 dB for each ICA component. We further checked visually the results for each selected ICA component (Fig. 4).

Overall we found that 45 components out of 115 (39%) were visible when compared to the preceding background (25 components belonged to the triggered-ICA and 20 to continuous ICA) and only 5 out of 115 (4%) (of them 4 were ICA

triggered) when compared at the moment of the markings (see Supplementary Data 1).

Statistical analysis was performed to study which factors could influence the visibility of the ICA component vs. the background. We used a generalized logistic mixed model in the R software, taking into account inter-patient variability, studying the variable "visibility" considered as a binary response (yes/no). We considered as factor influencing visibility: number of markers, markers location, network type, filtering, SEEG-triggered analysis, lateralization, number of structures implicated in the network.

**Code availability**. The code used to generate the results that are reported in this study is available from the corresponding authors upon reasonable request.

**Reporting summary**. Further information on experimental design is available in the Nature Research Reporting Summary linked to this article.

## Data availability
All data supporting the findings of this study are available from the corresponding authors upon reasonable request.

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

## Acknowledgements

This study was sponsored by Assistance Publique des Hôpitaux de Marseille (AP-HM). This work has been carried out within the FHU EPINEXT with the support of the A*MIDEX project (ANR-11-IDEX-0001–02) funded by the "Investissements d'Avenir" French Governement program managed by the French National Research Agency (ANR) Part of this work was funded by a joint Agence Nationale de la Recherche (ANR) and Direction Génerale de l'Offre de Santé (DGOS) under grant "VIBRATIONS" ANR-13-PRTS-0011-01. Part of this work was funded by a TechSan grant from Agence Nationale de la recherche "FORCE" ANR-13-TECS-0013. This work was performed on a platform member of France Life Imaging network (grant ANR-11-INBS-0006). Research supported by grants ANR-16-CONV-0002 (ILCB), ANR-11-LABX-0036 (BLRI) and the Excellence Initiative of Aix-Marseille University (A*MIDEX). Part of this work has been carried out thanks to the support of A*MIDEX (ANR-11-IDEX-0001–02 grant) funded by the French Government « Investissements d'Avenir » program. We thank Bruno Colombet for technical support and for designing the software Anyware. We thank all patients giving their consent for this study and the clinical staff for their support.

## Author contributions

F.P., N.R., S.M.V., J.M.B., F.B. and C.G.B. took part in the conception and design of the study. F.P., N.R., S.M.V., J.M.B., B.G., A.T. and C.G.B. contributed to data analysis. B.G., S. M.V. and N.R. performed all the statistical analysis. J.M.B., S.C. performed all the simultaneous recordings sessions. R.C. performed electrode implantation surgery. A.T., S.L., F.B., R.C., M.G. and A.M.G. contributed to manuscript, the design of the study and they were the referents for the patient's medical care.

## Additional information

**Competing interests:** The authors declare no competing interests.

