## [Peer Review File · Nature Communications]

Reviewers' comments:

Reviewer #1 (Remarks to the Author):

This manuscript presents data that support the hypothesis that MEG can record epileptiform activity (interictal spikes) that originate in the hippocampus and amygdala. The clinical and research potential for surface recordings from deep sources is great. My main concern with this paper is that it does so selecting from a large number of iEEG / MEG channels and ICA components to make the argument, but it is not clear how many events remain undetected, nor does it explore why that might be the case. Why do a large subset of patients do not show any correlations between iEEG electrodes and MEG sensors. While the basic finding of correlation between some channels in certain patients is interesting from a methodological standpoint, the orientation of the paper is largely descriptive and may be more suited for a more specialized audience.

PS: Figures need units

Reviewer #2 (Remarks to the Author):

This is an interesting study in which the authors state they are able to record activity that originates in the hippocampus or amygdala in the MEG signal recorded from an array of sensors placed around the surface of the head. They do this by recording simultaneous stereo EEG and MEG. The investigators perform ICA on the signal, at first triggered on focal epileptiform discharges in the amygdala and/ or hippocampus and then with free-running MEG/ SEEG. The authors state that analyzing certain ICA components demonstrates high correlation with specific activity recorded by SEEG depth electrodes in deep structures. They then conclude that these same techniques can be used to identify components of brain activity generated by these deep structures in "surface" MEG. The authors extend this conclusion by stating that they have identified numerous ICA components derived from analyzing the MEG signal that correspond to activity generated by networks in a number of different brain locations. They suggest that appropriately processed MEG may be quite useful in localizing and understand activity from deep brain structures that have classically been inaccessible to noninvasive neural recording. They state that using this technique could have a large number of applications in cognitive and other neuroscience research.

The article is clearly written and interesting. The presentation of the techniques is clear. Figures are a little redundant, showing magnetic field components mapped to MEG surface sensors that temporally correspond to activity recorded in the depth, and the manuscript could benefit from reducing the number of these.

A key question in this research is what exactly are the investigators looking at when they identify these MEG components by ICA? ICA is a technique that decomposes a signal into orthogonal components that contribute to the power of a particular signal. These individual components may have a specific mechanistic meaning, such as activity in a particular frequency, or connected with a particular temporal behavior. There is no requirement that they be associated with a particular spatial generator. It is possible that this could be the case, but this does not seem extremely likely. Stereo EEG techniques are based upon the concept of an epileptic network, where structures that generate and propagate epileptiform activity are connected to each other, are sampled, and regions critical to seizure generation are identified for removal, ablation or device placement.

Keeping this in mind, the question of causality comes forward. Are the signals that are seen in the ICA-filtered MEG generated by the regions being sampled by the MEG, or are they just conducting them or coherent with them? It is well known to epileptologists that epileptiform activity seen in the mesial temporal structures can be generated there or in other connected regions and then propagate there. The issue then becomes how can one tell if what is seen in the ICA-filtered MEG

is propagating to the sensors through the network, or reflected in the MEG because of dipole conduction through the CSF, which is in communication with the mesial temporal structures. Perhaps more likely is that the activity seen in the MEG that correlates with that in deep structures is part of a large, distributed limbic network that projects or connects to some of the structures imaged by the MEG. The authors mention using a "zero-lag" technique, which perhaps may eliminate signals that have a delay from the stereo EEG to the MEG, but this is not well explained.

The fact that the authors can capture signal components on the MEG that correlate with those in the depth Stereo EEG is still interesting, regardless of the mechanism of conduction. The worrisome part is the author's conclusion that the activity measured on the ICA-filtered MEG is generated by the SEEG sampled regions, rather than it sampling participation in whatever network is being imaged. This ambiguity detracts significantly from the conclusions of the paper. The manuscript almost seems to confound the concept of ICA and spatial generators at times.

There are some potential ways of getting at this issue. One is to stimulate or put low amplitude pulses, irregularly, into the amygdala and hippocampus, with amplitudes comparable to average interictal spikes recorded on the MEG, if this is possible, and verify that this activity is indeed seen on the ICA-filtered MEG. Another, better approach would be to take the components recorded on the MEG that correlate with deep activity, and use a dipole modeling algorithm or similar method on the data to see if this localizes to the region the investigators think it is coming from. There are certainly problems with dipole localization algorithms applied in this inverse solution manner, but it may be possible to constrain the solution to get a reasonable answer.

Overall the paper is interesting and aims to solve a problem with MEG that has plagued this technique since its original application to epilepsy almost 30 years ago- which is that MEG does not do a good job of localizing deep sources far removed from the surface sensors because the magnetic field falls off proportional to $1/r^2$. Claims of good localization of deep activity side tracked MEG investigators for many years, and it would be important not to replicate these difficulties. Certainly, a promising method that would allow investigators to record noninvasively from deep structures, like the hippocampus, amygdala, and other parts of the limbic network would solve a huge problem for researchers and clinicians. The issue of causality is the main hurdle for this paper, and given the importance of the question, it is vital that this issue be addressed definitively. I have given a couple of suggestions for ways to do this, but would be open to others.

Reviewer #3 (Remarks to the Author):

The paper deals with the important question of the capability of MEG to record activity from mesial temporal structures. This is a still debated issue that merits careful validation studies. The simultaneous recording of MEG and intracranial EEG (iEEG) provides a unique possibility to study this question. The authors used this methodology in 14 patients who underwent intracranial recordings in the framework of presurgical epilepsy evaluation.

MEG was analysed using independent component analysis and the time course of the ICA components were correlated with the intracranial recordings. Specific ICA maps were found for purely mesial temporal (hippocampus or amygdala) or even deeper (thalamus) activity in selected patients, indicating that such activity is visible with MEG.

The problem with this finding is that it is based on topographic analysis on the sensor level only. As correctly pointed out by the authors and demonstrated in Figure 2, mesial sources can extend to lateral sources even at zero lag. The bipolar SEEG recordings only sparsely sample the lateral cortex and it can thus not be excluded that lateral cortical activity was present but not captured by the implanted electrode contacts.

Topographic maps that are specific to mesial temporal activity have been demonstrated previously. Already in the 1990ies, John Ebersole showed that specific EEG voltage map topographies (and their modeled dipole sources) represent mesial temporal spikes (Ebersole,

Epilepsia 2000). Later, combined EEG-fMRI studies demonstrated that the time-course of spike-specific EEG maps correlated with BOLD activity in the focus, also in mesial temporal lobe epilepsy (Grouiller et al., Brain, 2011). Thus, the demonstration of specific topographic maps on the scalp that correlate with mesial temporal activity is not new, but not sufficient to claim that it is the mesial activity only that produced the map.

What is missing this study is the source localization of these components and the demonstration that the solution points closest to the intracranial electrode indeed show highest zero-lag correlation. As the data are presented now, lateral sources contributing to the map (but not sampled by the intracranial electrodes) cannot be excluded.

Reviewer #1 (Remarks to the Author):

My main concern with this paper is that it does so selecting from a large number of iEEG / MEG channels and ICA components to make the argument, but it is not clear how many events remain undetected, nor does it explore why that might be the case. Why do a large subset of patients do not show any correlations between iEEG electrodes and MEG sensors. While the basic finding of correlation between some channels in certain patients is interesting from a methodological standpoint, the orientation of the paper is largely descriptive and may be more suited for a more specialized audience.

In Table 2, we report for all patients the number of events marked on SEEG and the different ICA components detected on MEG. As pointed out by the reviewer, some patients (i.e. P13; P14) did not show mesial components on MEG analysis despite the high number of spikes marked on the SEEG.

In order to explain the lack of detection in some cases, different hypotheses were tested statistically: number of events, network type, filtering, brain structure. We found that neither the number of marked events nor the extension of the network influenced detectability. However, significant effects were observed with filter band and marked structure (hippocampus versus amygdala). We hypothesize that other factors that are difficult to evaluate, such as the orientation of the active cortex or the state of the patient during the recording, could play a role in detectability. These results and hypotheses are presented in the discussion part of the manuscript.

PS: Figures need units

We provided units to figures. Correlation values and the amplitude of the ICA component and SEEG correlated signals are in arbitrary units (a.u.).

Reviewer #2 (Remarks to the Author):

Figures are a little redundant, showing magnetic field components mapped to MEG surface sensors that temporally correspond to activity recorded in the depth, and the manuscript could benefit from reducing the number of these.

In the revised version we removed two figures (corresponding to Fig 2 and part of Fig 3 in the original submission)

A key question in this research is what exactly are the investigators looking at when they identify these MEG components by ICA?... Are the signals that are seen in the ICA-filtered MEG generated by the regions being sampled by the MEG, or are they just conducting them or coherent with them? ... Perhaps more likely is that the activity seen in the MEG that correlates with that in deep structures is part of a large, distributed limbic network that projects or connects to some of the structures imaged by the MEG.

We fully agree with the reviewer; this is in fact the key issue that we wanted to address in the study. Our hypothesis was that independent component analysis is able to disentangle local

activity (within mesial structures) from propagated activity (within large networks). We indeed found (within the same patients, ex: P5) different ICA components, some correlating with only one mesial structure (Tab 2, Fig 1), other with multiple mesio-lateral structures (Tab 2), which is in line with our hypothesis.

We have done our best, using two different types of correlations and evaluating presence of zero-lag synchrony, to ensure that the activity on MEG actually reflects direct mesial activity and not propagated activity through neuronal networks. It is important to note that the SEEG sampling is tailored to sample the regions hypothesized to be epileptogenic and that the lateral regions, (which are often regions of propagation of mesial activity) are sampled by lateral contacts of the SEEG electrodes. As epileptic activity often involves large areas of cortex, we therefore think that the probability of missing important propagated activity is reasonably low (but not null, of course, see response below).

In the revised manuscript, we added a paragraph in the beginning of the results in order to better explain the rationale of the correlation/zero lag method. We also added a presentation of the SEEG implantation methodology and the mesiolateral sampling (see revised discussion). In addition, we have added numbers of electrodes and contacts recorded in our cohort (methods part).

The authors mention using a "zero-lag" technique, which perhaps may eliminate signals that have a delay from the stereo EEG to the MEG, but this is not well explained.

We now added to the manuscript a description on zero-lag analysis in the context of possible signal propagation (cf Results - first paragraph)

The worrisome part is the author's conclusion that the activity measured on the ICA-filtered MEG is generated by the SEEG sampled regions, rather than it sampling participation in whatever network is being imaged (..) There are some potential ways of getting at this issue. (..) A better approach would be to take the components recorded on the MEG that correlate with deep activity, and use a dipole modeling algorithm or similar method on the data to see if this localizes to the region the investigators think it is coming from.

We fully agree that, despite our precautions, and despite the high spatial sampling provided by our SEEG methodology (up to 19 electrodes and 249 contacts in a single patient), it is not possible to exclude the observation of activity propagated from the mesial structures to other (remote) structures not sampled by SEEG. Following the reviewer's advice, we performed source analysis on the ICA components. To our knowledge, we are stricter in our criteria (regarding local versus propagation) than any previous study correlating depth and surface measures.

We confirmed in a vast majority of cases (8 out of 9 for "SEEG- triggered" analysis, and 9 out of 14 for "continuous" analysis) that the source of the ICA components of interest was indeed located mesially. Due to the uncertainty in locating deep activity (see for example Hati et al, *Electroenceph and Clin Neurophysiol*, 1988), we think it is difficult to ensure that the source is located precisely within the structure of interest. Still, we computed a confidence interval of

the dipole source, and found that in 78 % of the cases for “SEEG- triggered” analysis and in 29% for “continuous” analysis the structure was included in the confidence interval. This suggests that MEG-only analysis has to be improved but does not hamper our main goal (i.e, to show that mesial activity does produce a signal visible on MEG sensors)

All these results are presented in the revised version of the manuscript. We are now confident that, after the multistep zero-lag temporal correlation method and source localization, the selected mesial components indeed reflect a mesio-temporal generator or in some cases even deeper sources.

Reviewer #3 (Remarks to the Author):

The problem with this finding is that it is based on topographic analysis on the sensor level only.... What is missing this study is the source localization of these components and the demonstration that the solution points closest to the intracranial electrode indeed show highest zero-lag correlation. As the data are presented now, lateral sources contributing to the map (but not sampled by the intracranial electrodes) cannot be excluded.

The possible contribution of lateral sources not sampled by SEEG on MEG mesial ICA component is indeed a critical point. Thanks to the widespread intracranial sampling (up to 19 SEEG electrodes for P13) we have a high spatial coverage of the lateral regions. However, as pointed out by the reviewer, the presence of other lateral sources far from the SEEG electrodes contributing to the ICA mesial map cannot theoretically be excluded.

As suggested by the reviewer, we performed source localization on the mesial ICA components. The topographies of ICA mesial components (“distant maxima” dipolar field) can be complex and difficult to localize, consistent with previous studies and reporting high localization error for deep sources (Hati et al, *Electroenceph Clin Neurophysiol* 1988).

We found in the majority of cases (8 out of 9 for “SEEG- triggered” analysis, and 9 out of 14 for “continuous” analysis), a confirmation of the mesial origin of the MEG ICA component using source localization. Despite the difficulty of precisely assigning the dipole to a given deep structure, we found that in 78 % of the cases for “SEEG- triggered” analysis and in 29% of cases for “continuous” analysis, the structure was within the confidence interval of the dipole source.

Details about source localization are reported in a specific paragraph in the Methods section of the revised version. Source localization was added to figures reported in the previous version and described in Table 2 and in the results part. We also added a supplementary Figure with a representation of source localization for each patient with a mesial ICA component.

Reviewers' comments:

Reviewer #1 (Remarks to the Author)

The authors have addressed my previous concerns

Reviewer #3 (Remarks to the Author):

On the request of the reviewers, the authors now provide source analysis of the ICA components using a single (or dual) dipole fit method. Out of the 115 ICA components that were retained after the correlation analysis, only those that were correlated with mesial or extended mesial SEEG contacts were source localized (a total of 41 out of the 115 components, i.e. 35%). Not all of these components were independent. Those selected from the 2-60 Hz filtered data were probably often the same as those selected from the 12-60 Hz filtered data, and those selected from the continuous and from the spike-aligned data were probably also often the same. Similarities between these components within a patient are not described.

Out of the 41 localized components, only 21 correctly localized in mesial structures that included the SEEG contact (51%). That means that from 115 selected ICA components, only 21 (18%) showed convincing localization in the mesial structures. Given that some of these components were probably the same as described above, this is a rather modest result.

What is missing is the localization of the non-mesial components and the demonstration that in these cases the dipole source was indeed not localized in the mesial structures. This would be a more convincing demonstration of the specificity of the ICA components for mesial sources. If the dipoles of many of the components labeled as "lateral" are also localized in the mesial structures, the validity of the dipole localization method as a proof for the visibility of mesial sources can be questioned.

Point by point answer to reviewer 3

Reviewer 3: On the request of the reviewers, the authors now provide source analysis of the ICA components using a single (or dual) dipole fit method. Out of the 115 ICA components that were retained after the correlation analysis, only those that were correlated with mesial or extended mesial SEEG contacts were source localized (a total of 41 out of the 115 components, i.e. 35%).

Response: In the revised version we added source localization of lateral ICA components (31 components, up to 62% of total). Localization of Mesio-lateral (ML) and extended limbic (eL) ICA localization was not performed due to inherent difficulties in fitting dipoles on such distributed networks.

Reviewer 3: Not all of these components were independent. Those selected from the 2-60 Hz filtered data were probably often the same as those selected from the 12-60 Hz filtered data, and those selected from the continuous and from the spike-aligned data were probably also often the same. Similarities between these components within a patient are not described.

We agree. Similarities between ICA components across the different types of analysis within a patient was quantified in the revised version using hierarchical clustering. Thus, clustering was applied to ICA topographies within each patient.

This analysis showed that the 115 components could be summarized by only 63 clusters. However, in the particular case of mesio-temporal ICA components (31), a high number of clusters were found (27). This confirms the importance of the different types of analysis for maximizing the possibility to retrieve mesio-temporal components.

The possibility of obtaining pure mesial ICA components in MEG is the main result of our study. As evidenced by clustering analysis, topographies similarities across the mesial ICA components of the same patient is low. Taken together, we think these new results do not impact the global message of the paper.

We added to the manuscript specific subsections within the Methods and Results sections, under "ICA component classification" and "ICA, independent of SEEG-markers, is also able to detect mesio-temporal sources" results' parts, as well as in the Discussion.

Reviewer 3: Out of the 41 localized components, only 21 correctly localized in mesial structures that included the SEEG contact (51%). That means that from 115 selected ICA components, only 21 (18%) showed convincing localization in the mesial structures. Given that some of these components were probably the same as described above, this is a rather modest result.

Response: Source localization of deep activity is a difficult issue where, to the best of our knowledge, no consensus exists on the best method to use. We added a specific paragraph on this topic to the Discussion. Among the difficulties are the biophysical source model, the low signal to noise ratio, and the fact that on a single topography one cannot estimate the noise covariance matrix. Thus, it is

important to note that in 8/41 of the mesial cases (34%) the goodness of fit (GOF) was too small to allow reliable dipolar source localization.

Within the 33 ICA mesial components with a good GOF, source localization was in the mesial brain regions in 27 cases (81%). In 21 cases source localization included the brain structure with the highest correlation value (63%). In sum, we found convincing deep activities in 11 patients out of 14 (correlation and source localization). In our opinion, these results are significant and useful, especially given the scarce general experience in source localization of mesial sources.

Reviewer 3: What is missing is the localization of the non-mesial components and the demonstration that in these cases the dipole source was indeed not localized in the mesial structures. This would be a more convincing demonstration of the specificity of the ICA components for mesial sources. If the dipoles of many of the components labeled as "lateral" are also localized in the mesial structures, the validity of the dipole localization method as a proof for the visibility of mesial sources can be questioned.

Response: In the revised version, we added the localization results for the lateral components and we investigated sensitivity and specificity of retrieving deep brain sources. Thus, in the cases with high GOF, we found a sensitivity of 0.81 (27/33) and a specificity of 0.93 (28/30). Methods and results are reported in the specific paragraph in the Methods section.

REVIEWERS' COMMENTS:

Reviewer #3 (Remarks to the Author):

The authors adequately answered to my comments. The added cluster analysis of the ICA components and the measure of sensitivity and specificity respond to my concerns and improve the quality of the paper.

I have one minor point that might deserve a consideration. It concerns Figure 2 and 3 and the discussion about the detection of thalamic sources:

Figure 2 shows an ICA component that correlates with both, hippocampal and thalamic activity, and source localization points to both structures. Since a single dipole localization method was used in most cases, it is not clear to me whether this single dipole GOF extends from thalamus to hippocampus or whether in this case a two dipole model was used. I cannot find a description of the source localization approach in this case.

Fig. 3 shows correlation with the thalamus only. However, in this case the correlation value seems to be much lower than in the other cases (the maximum in the inset is 0.14). Given that the significance test for the correlations was based on IFDR with a threshold set at 0.2, I was wondering whether the correlation in this case was indeed significant.

Answer to the reviewer :

“The authors adequately answered to my comments. The added cluster analysis of the ICA components and the measure of sensitivity and specificity respond to my concerns and improve the quality of the paper.

I have one minor point that might deserve a consideration. It concerns Figure 2 and 3 and the discussion about the detection of thalamic sources:

Figure 2 shows an ICA component that correlates with both, hippocampal and thalamic activity, and source localization points to both structures. Since a single dipole localization method was used in most cases, it is not clear to me whether this single dipole GOF extends from thalamus to hippocampus or whether in this case a two dipole model was used. I cannot find a description of the source localization approach in this case.”

- We specified in the figure legends the type of dipole fitting that was used (in these cases, single dipoles)

“Fig. 3 shows correlation with the thalamus only. However, in this case the correlation value seems to be much lower than in the other cases (the maximum in the inset is 0.14). Given that the significance test for the correlations was based on IFDR with a threshold set at 0.2, I was wondering whether the correlation in this case was indeed significant.”

- The IFDR is an empirical Bayesian method that estimates the threshold based on the histogram of all values (in order to take into account multiple comparison). It boils down to assessing whether a values ‘stands out’ of the noise. The 0.2 is the ‘local’ threshold on the ratio between the estimated H_0 and total number of detections, not the threshold on the original correlation values (in other terms, 0.2 is the equivalent for IFDR of the $Q=0.5$ in “classical” FDR). In the revised version, we explained these issues in the methods section.

Importantly, we always measure in addition to correlation across time the correlation across repetitions of the spikes (for each time point, ITCOR). In this case, the peak of correlation was 0.59. We only kept findings where both correlation across time and across repetitions are significant.